# THE RESPONSIBILITY PROBLEM IN NEURAL NETWORKS WITH UNORDERED TARGETS

**Ben Hayes, Charalampos Saitis & György Fazekas**
School of Electronic Engineering and Computer Science
Queen Mary University of London, UK
{b.j.hayes,c.saitis,george.fazekas}@qmul.ac.uk

## ABSTRACT

We discuss the discontinuities that arise when mapping unordered objects to neural network outputs of fixed permutation, referred to as the *responsibility problem*. Prior work has proved the existence of the issue by identifying a single discontinuity. Here, we show that discontinuities under such models are uncountably infinite, motivating further research into neural networks for unordered data.

## 1 INTRODUCTION

The *responsibility problem* (Zhang et al., 2020b) describes an issue when training neural networks with unordered targets: the fixed permutation of output units requires that each assume a "responsibility" for some element. Permutations of responsibility result in discontinuities wherein small changes in a permutation invariant metric demand large changes in the layer's actual output. For feed-forward networks, the worst-case approximation of such discontinuous functions is arbitrarily poor for at least some subset of the input space (Kratsios & Zamanlooy, 2022)

Empirically, degraded performance has been observed on set prediction tasks (Zhang et al., 2020a), motivating research into architectures for set generation which circumvent these discontinuities (Zhang et al., 2020a; Kosiorek et al., 2020; Rezatofighi et al., 2018). The problem has briefly been stated formally in the literature (Zhang et al., 2020b), but proven only by the existence of a single discontinuity. Here we prove an alternative statement of the responsibility problem, categorising unordered-to-ordered mappings as operations that are isomorphic to sorting and operations that are not.

In the first case, the discontinuities that arise are "classic" permutations of responsibility – that is, an element that was initially represented by one output unit becomes represented by another. Using a result from Hemasinha & Weaver (2015), we find that any mappings that preserve a consistent ordering between elements are isomorphic and thus all admit such discontinuities. In the second case, discontinuities also appear wherever the map does *not* implement a consistent ordering. We demonstrate that, as a consequence, the set of discontinuities in both sorting and non-sorting maps is uncountably infinite.

This result is congruent with experimental evidence that performance suffers when the orderless structure of set data is not explicitly accounted for in model architectures. This has practical implications for any task that inherently involves set prediction, including object detection, point cloud generation, molecular graph generation, speech separation, and more.

## 2 THE RESPONSIBILITY PROBLEM

Let $\Theta$ denote the set of sets $\boldsymbol{\theta}$ of cardinality $|\boldsymbol{\theta}| = n$ with elements in $\mathbb{R}^d$ where $d \geq 2$:

$$\boldsymbol{\theta} \triangleq \left\{ \boldsymbol{x}_k \mid \boldsymbol{x}_k \in \mathbb{R}^d, k \in \{1, \ldots, n\} \right\} \in \Theta.$$

We are interested in the set $\mathcal{F}$ of maps $f \colon \Theta \to \mathbb{R}^{n \times d}$, such that all $f \in \mathcal{F}$ select a single assignment of each element in any $\boldsymbol{\theta} \in \Theta$ to each row of the resulting matrix:

$$f(\boldsymbol{\theta}) = \begin{bmatrix} \boldsymbol{x}_{\pi(1)} & \boldsymbol{x}_{\pi(2)} & \cdots & \boldsymbol{x}_{\pi(n)} \end{bmatrix}^T, \qquad \forall f \in \mathcal{F}, \quad \forall \boldsymbol{\theta} \in \Theta, \quad \pi = \Pi_f(\boldsymbol{\theta}) \ ,$$

where $\Pi_f \colon \Theta \to S_n$ describes the permutation applied to the indices of $\boldsymbol{\theta}$ and $S_n$ is the symmetric group of order $n$. Such a map represents the assignment from an unordered object, such as a set, to the units of a neural network layer. We thus wish to show that all $f \in \mathcal{F}$ must be discontinuous, proceeding first with some definitions.

**Definition 1.** *We call $\Pi_f \colon \Theta \to S_n$ a sorting operation under an order $\leq$ if for all $\boldsymbol{\theta} \in \Theta$:*

$$\boldsymbol{x}_{\pi(1)} \leq \boldsymbol{x}_{\pi(2)} \leq \cdots \leq \boldsymbol{x}_{\pi(n)}, \quad \pi = \Pi_f(\boldsymbol{\theta}).$$

Let $\mathcal{V}$ be an ordered vector space consisting of $\mathbb{R}^d$ equipped with a total order denoted $\ll$. That is, $\mathcal{V} = (\mathbb{R}^d, \ll)$.

**Definition 2.** *A total order $\ll$ over $\mathcal{V}$ is called lexicographic if:*

1. $\boldsymbol{x} = \boldsymbol{y} \implies \boldsymbol{x} \ll \boldsymbol{y}$                                 $\forall \boldsymbol{x}, \boldsymbol{y} \in \mathcal{V}$
2. $\boldsymbol{x} \ll \boldsymbol{y} \wedge \boldsymbol{x} \neq \boldsymbol{y} \implies x_i < y_i$      $i = \min\{j | x_j \neq y_j\}, \forall \boldsymbol{x}, \boldsymbol{y} \in \mathcal{V}$ .

**Definition 3.** *A basis set $\{\boldsymbol{v}_i\}_{i \in \{1,\ldots,d\}}$ for $\mathcal{V}$ is called non-Archimedian if $0 \ll \boldsymbol{v}_d \ll \cdots \ll \boldsymbol{v}_1$ and $a\boldsymbol{v}_{i+1} \ll \boldsymbol{v}_i, \forall a \in \mathbb{R}$.*

Then we recall the following theorem from Hemasinha & Weaver (2015).

**Theorem 1.** *Any total order $\ll$ over $\mathbb{R}^d$ where $d \geq 2$ is isomorphic to the lexicographic ordering on a non-Archimedean basis.*

We can hence continue under the assumption that $\mathcal{V}$ is defined on a non-Archimedian basis under the lexicographic order $\ll$, without loss of generality. Then, we can prove the following lemma (proof given in Appendix A).

**Lemma 1.** *If $\Pi_f$ is not a sorting operation under a total order, then $\Pi_f$ is not a sorting operation at all.*

One of the following statements is therefore true about a given $\Pi_f$: (1) $\Pi_f$ is isomorphic to a sorting operation under the lexicographic order, or (2) $\Pi_f$ is not a sorting operation.

This leads to the following lemma for discontinuity in the second case (proof in Appendix A.1).

**Lemma 2.** *For a map $f \in \mathcal{F}$, if $\Pi_f$ is not a sorting operation, then $f$ is discontinuous.*

Therefore, we focus now only on the case where $\Pi_f$ is a sorting operation under $\ll$, giving the following lemma (proven in Appendix A.1).

**Lemma 3.** *For a map $f \in \mathcal{F}$, if $\Pi_f$ is a sorting operation, then $f$ is discontinuous.*

By exhaustion, the proof of the responsibility problem is then trivial.

**Theorem 2.** *All $f \in \mathcal{F}$ are discontinuous.*

*Proof.* By Lemmas 2 and 3 we have:

$$(f \text{ is a sorting operation}) \vee (f \text{ is not a sorting operation}) \implies f \text{ is discontinuous}, \quad \forall f \in \mathcal{F}$$

The antecedent is clearly a tautology, so all $f \in \mathcal{F}$ are discontinuous, and we have proven the existence of the responsibility problem. $\square$

Moreover, having proved Theorem 2 in terms of sorting and non-sorting maps yields the following corollary (proof in Appendix A.1).

**Corollary 1.** *For a map $f \in \mathcal{F}$, the set of $\theta \in \Theta$ for which $f$ is discontinuous is uncountably infinite.*

## 3   CONCLUSION

We have provided an alternate statement of the responsibility problem in neural networks, and have for the first time proven its existence for an uncountably infinite class of discontinuity. The existence and scope of this problem motivates further research into neural network architectures for set generation and into the behaviour of common optimization algorithms in the presence of such discontinuities.

ACKNOWLEDGEMENTS

We thank Louise Thorpe and Shubhr Singh for their helpful and insightful feedback.

URM STATEMENT

The authors acknowledge that at least one key author of this work meets the URM criteria of ICLR 2023 Tiny Papers Track.

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

## A PROOFS

**Lemma 1.** *If $\Pi_f$ is not a sorting operation under a total order, then $\Pi_f$ is not a sorting operation at all.*

*Proof.* Let us say that $\Pi_f$ is a sorting operation under an order denoted $\lhd$. Then $\lhd$ is necessarily a total order because,

$$f(\boldsymbol{\theta}) = \begin{bmatrix} \boldsymbol{x}_{\pi(1)} & \boldsymbol{x}_{\pi(2)} & \dots & \boldsymbol{x}_{\pi(n)} \end{bmatrix} \iff \begin{array}{c} \boldsymbol{x}_{\pi(1)} \lhd \boldsymbol{x}_{\pi(2)} \lhd \cdots \lhd \boldsymbol{x}_{\pi(n)}, \\ \pi = \Pi_f(\boldsymbol{\theta}), \ \forall \boldsymbol{\theta} \in \Theta. \end{array}$$

That is, the binary operation $\lhd$ must be defined for all $\boldsymbol{x} \in \mathbb{R}^d$ and is therefore a total order by the totality axiom. $\square$

### A.1 MAIN PROOFS

To proceed with proving Lemmas 2 and 3, we define $d_\Theta \colon \Theta \times \Theta \to \mathbb{R}$ as a metric of the form:

$$d_\Theta(\boldsymbol{\theta}, \boldsymbol{\theta}') = \left\| \sum_{\boldsymbol{x} \in \boldsymbol{\theta}} \phi(\boldsymbol{x}) - \sum_{\boldsymbol{x}' \in \boldsymbol{\theta}'} \phi(\boldsymbol{x}') \right\|_p,$$

where $\phi \colon \mathbb{R}^d \to \mathbb{R}^k$ is a continuous map from elements in $\boldsymbol{\theta}$ to an arbitrary representation. $(\Theta, d_\Theta)$ is then a metric space.

**Lemma 2.** *For a map $f \in \mathcal{F}$, if $\Pi_f$ is not a sorting operation, then $f$ is discontinuous.*

*Proof.* We wish to show that under the Frobenius norm $\|\cdot\|_F$ over $\mathbb{R}^{n \times d}$, there exists an $\epsilon > 0$ such that for all $\delta > 0$ there exist two sets $\boldsymbol{\theta}, \boldsymbol{\theta}' \in \Theta$ such that $d_\Theta(\boldsymbol{\theta}, \boldsymbol{\theta}') < \delta$ and $\|f(\boldsymbol{\theta}) - f(\boldsymbol{\theta}')\|_F \geq \epsilon$.

Assume $\Pi_f$ is not a sorting operation for some $f \in \mathcal{F}$, then there must exist at least one pair of elements in $\mathcal{V}$ for which $\Pi_f$ violates Definition 1. It follows, then, that there exists $\boldsymbol{a}, \boldsymbol{b} \in \mathcal{V}$ with $\boldsymbol{a} \ll \boldsymbol{b}$ and $\boldsymbol{a} \neq \boldsymbol{b}$ such that there exist $\boldsymbol{\theta}, \boldsymbol{\theta}' \in \Theta$ defined:

$$
\begin{aligned}
\boldsymbol{\theta} &= \{\boldsymbol{a}, \boldsymbol{b}, \boldsymbol{u}_1, \boldsymbol{u}_2, \ldots, \boldsymbol{u}_{n-2}\} \\
\boldsymbol{\theta}' &= \{\boldsymbol{a}, \boldsymbol{b}, \boldsymbol{u}_1', \boldsymbol{u}_2', \ldots, \boldsymbol{u}_{n-2}'\}
\end{aligned} ,
$$

where the permutations produced by $f$ assume the following form:

$$
\begin{aligned}
f(\boldsymbol{\theta}) &= [\ldots \quad \boldsymbol{a} \quad \ldots \quad \boldsymbol{b} \quad \ldots]^T \\
f(\boldsymbol{\theta}') &= [\ldots \quad \boldsymbol{b} \quad \ldots \quad \boldsymbol{a} \quad \ldots]^T ,
\end{aligned}
$$

where the indices of the rows of $f(\boldsymbol{\theta})$ corresponding to $\boldsymbol{a}, \boldsymbol{b}$ do not necessarily equal the indices of the rows of $f(\boldsymbol{\theta}')$ corresponding to $\boldsymbol{a}, \boldsymbol{b}$. The existence of such a pair is necessary because if $\ll$ is inverted for for all $\boldsymbol{\theta} \in \Theta$ containing $\boldsymbol{a}, \boldsymbol{b}$, then the result is isomorphic to $\ll$ up to renaming of elements. By this necessary inversion of ordering, we can form the following inequality because only one of $\boldsymbol{a}$ or $\boldsymbol{b}$ can possibly be assigned to the same column in both matrices:

$$
\|f(\boldsymbol{\theta}) - f(\boldsymbol{\theta}')\|_F^2 \geq \min\left\{ \min_{i,j} \|\boldsymbol{a} - \boldsymbol{u}_i\|_2^2 + \|\boldsymbol{a} - \boldsymbol{u}_j'\|_2^2, \min_{i,j} \|\boldsymbol{b} - \boldsymbol{u}_i\|_2^2 + \|\boldsymbol{b} - \boldsymbol{u}_j'\|_2^2 \right\}.
$$

Let this quantity be $\epsilon^2$. Let $A \subset \Theta$ be the set of all $\boldsymbol{\theta}$ containing $\boldsymbol{a}, \boldsymbol{b}$ where Definition 1 holds for $\boldsymbol{a}$ and $\boldsymbol{b}$, and let $B \subset \Theta$ be the set of all $\boldsymbol{\theta}'$ containing $\boldsymbol{a}, \boldsymbol{b}$ where Definition 1 does not hold. Let $C = A \cup B$. Then $C$ is a metric space under $d_\Theta$, with $A, B \subset C$ and $A = B^c$.

There must therefore exist a sequence $(\boldsymbol{\theta}_m)_{m \in \mathbb{N}}$ in $A$ and an element $\boldsymbol{\theta}' \in \partial B$ where $\lim_{m \to \infty} \boldsymbol{\theta}_k = \boldsymbol{\theta}'$. Hence we can find $\boldsymbol{\theta} \in A, \boldsymbol{\theta}' \in B$ such that $d_\Theta(\boldsymbol{\theta}, \boldsymbol{\theta}') < \delta$ for all $\delta > 0$.

Having defined $\epsilon$, we have thus satisfied the discontinuity criterion. $\square$

To prove Lemma 3, we require a simple intermediate result.

**Lemma 4.** *All maps $f \in \mathcal{F}$ are injective.*

*Proof.* Assume there is a map $g \in \mathcal{F}$ that is not injective. Then we have:

$$
\exists \boldsymbol{\theta}, \boldsymbol{\theta}' \in \Theta : \qquad f(\boldsymbol{\theta}) = f(\boldsymbol{\theta}') \wedge \boldsymbol{\theta} \neq \boldsymbol{\theta}'.
$$

However, by definition the set of rows of the matrix $\{f(\boldsymbol{\theta})_i \mid i \in \{1, \ldots, n\}\}$ is equivalent to the set $\boldsymbol{\theta}$, giving:

$$
\boldsymbol{\theta} = \{f(\boldsymbol{\theta})_i \mid i \in \{1, \ldots, n\}\} = \{f(\boldsymbol{\theta}')_i \mid i \in \{1, \ldots, n\}\} = \boldsymbol{\theta}' \neq \boldsymbol{\theta},
$$

which is a contradiction. $\square$

**Lemma 3.** *For a map $f \in \mathcal{F}$, if $\Pi_f$ is a sorting operation, then $f$ is discontinuous.*

*Proof.* We again wish to show that under the Frobenius norm $\|\cdot\|_F$ over $\mathbb{R}^{n \times d}$, there exists an $\epsilon > 0$ such that for all $\delta > 0$ there exist two sets $\boldsymbol{\theta}, \boldsymbol{\theta}' \in \Theta$ such that $d_\Theta(\boldsymbol{\theta}, \boldsymbol{\theta}') < \delta$ and $\|f(\boldsymbol{\theta}) - f(\boldsymbol{\theta}')\|_F \geq \epsilon$.

Consider two vectors $\boldsymbol{a}, \boldsymbol{b} \in \mathcal{V}$ such that $\boldsymbol{b} = \boldsymbol{a} + \epsilon \boldsymbol{v}_j$ for $\epsilon > 0$ and for some $j > 1$, where $\{\boldsymbol{v}_i\}_{i \in \{1, \ldots, d\}}$ is the non-Archimedean basis of $\mathcal{V}$. We then have that $\boldsymbol{a} \ll \boldsymbol{b}$ by definition of the lexicographic order. It thus follows that the matrix $\boldsymbol{A}$ is in the image of $\Theta$ under $f$:

$$
\boldsymbol{A} = [\ldots \quad \boldsymbol{a} \quad \boldsymbol{b} \quad \ldots]^T \in f[\Theta].
$$

By injectivity of $f$ there is a unique $\boldsymbol{\theta} \in \Theta$ that maps to $\boldsymbol{A}$:

$$
\boldsymbol{A} = f(\boldsymbol{\theta}) \implies \boldsymbol{\theta} = \{\ldots, \boldsymbol{a}, \boldsymbol{b}, \ldots\}.
$$

Let $B_{\delta'}(\boldsymbol{b})$ be the open ball of radius $\delta' > 0$ and centre $\boldsymbol{b}$. Let $N_{\boldsymbol{a}} \triangleq \{\boldsymbol{v} \mid \boldsymbol{v} \in \mathcal{V}, \boldsymbol{v} \ll \boldsymbol{a}, \boldsymbol{v} \neq \boldsymbol{a}\}$ be the set of all elements in $\mathcal{V}$ that are strictly less than $\boldsymbol{a}$ under $\ll$. We wish to find an element $\boldsymbol{c} \in B_{\delta'}(\boldsymbol{b}) \cap N_{\boldsymbol{a}}$. In other words, we seek, an element in the $\delta'$-ball around $\boldsymbol{b}$ that is strictly less than $\boldsymbol{a}$ under the lexicographic order $\ll$. We can find such an element by taking $\boldsymbol{c} = \boldsymbol{b} - \tau \boldsymbol{v}_1$ for $0 < \tau < \delta'$.

Then another matrix $\boldsymbol{A}' = f(\boldsymbol{\theta}')$ where $\boldsymbol{\theta}' = \{\ldots, \boldsymbol{a}, \boldsymbol{c}, \ldots\}$, assuming all unspecified elements equal to those in $\boldsymbol{\theta}$, must take the form:

$$\boldsymbol{A}' = [\ldots \quad \boldsymbol{c} \quad \boldsymbol{a} \quad \ldots]^T = f(\boldsymbol{\theta}').$$

Then we have:

$$\|f(\boldsymbol{\theta}) - f(\boldsymbol{\theta}')\|_F = \|\boldsymbol{A} - \boldsymbol{A}'\|_F = \sqrt{\|\boldsymbol{a} - \boldsymbol{c}\|_2^2 + \|\boldsymbol{b} - \boldsymbol{a}\|_2^2 + \ldots} \geq \|\boldsymbol{b} - \boldsymbol{a}\|_2 = \epsilon.$$

By definition of $d_\Theta$ we have:

$$d_\Theta(\boldsymbol{\theta}, \boldsymbol{\theta}') = \|\phi(\boldsymbol{b}) - \phi(\boldsymbol{c})\|_p.$$

By continuity of $\phi$ we can always select a $\delta'$ such that $d_\Theta(\boldsymbol{\theta}, \boldsymbol{\theta}') < \delta$ for arbitrarily small $\delta$. Hence, we have found a class of $\boldsymbol{\theta} \in \Theta$ at which $f$ is discontinuous. $\qquad \square$

**Corollary 1.** *For a map $f \in \mathcal{F}$, the set of $\theta \in \Theta$ for which $f$ is discontinuous is uncountably infinite.*

*Proof.* First assume that $\Pi_f$ is a sorting operation under $\ll$. In the proof of Lemma 3, we selected a value for $\boldsymbol{a} \in \mathbb{R}^d$ arbitrarily. It follows that for any choice of $\boldsymbol{a} \in \mathbb{R}^d$ we can find $\boldsymbol{\theta}$ at which $f$ has a discontinuity. Let $D_f$ denote the set of $\boldsymbol{\theta}$ at which $f$ is discontinuous, then we have $|D_f| \geq |\mathbb{R}^d| > \aleph_0$, so $D_f$ must be uncountable.

In the case where $\Pi_f$ is not a sorting operation under $\ll$, we can partition $\Theta$ into two subsets: $\Theta_s$ which contains all $\boldsymbol{\theta}$ that $\Pi_f$ sorts according to $\ll$, and $\Theta_u$ which contains all $\boldsymbol{\theta}$ that $\Pi_f$ does not sort according to $\ll$.

We then follow the construction in the proof of Lemma 3 to find $\boldsymbol{\theta}, \boldsymbol{\theta}'$. Then either (i) both $\boldsymbol{\theta}$ and $\boldsymbol{\theta}'$ are in $\Theta_s$, in which case we have a discontinuity at $\boldsymbol{\theta}$ by Lemma 3 or (ii) at least one of $\boldsymbol{\theta}$ or $\boldsymbol{\theta}'$ is in $\Theta_u$, in which case we have a discontinuity by Lemma 2. One of these cases must hold for all choices of $\boldsymbol{a} \in \mathbb{R}^d$, so again we have $|D_f| \geq |\mathbb{R}^d| > \aleph_0$, so $D_f$ must be uncountable for all $f \in \mathcal{F}$. $\qquad \square$

