# OpenReview forum: "The Responsibility Problem in Neural Networks with Unordered Targets"
_ICLR.cc/2023/TinyPapers — Submitted to Tiny Papers @ ICLR 2023_

### Official Review · Reviewer_L2So · 2023-03-23

**Confidence:** 1

**Summary Of Contributions:**

This paper proves that maps which assign set elements to a fixed ordering must be discontinuous.

**Rating:**

Great Start (GS): a submission which meets some of the reviewing criteria but has room for improvement

**Strengths And Weaknesses:**

I assess the paper's strengths and weaknesses with respect to each of the reviewing criteria below:

- **Clarity**: The text is fairly well-organized, and at a high level its logical structure is easy to understand.

- **Correctness**: I have not checked the proofs carefully, because I am not familiar with many of the relevant references.

- **Reproducibility**: To the best of my understanding, adequate details are given in the main text and appendices.

- **Follows basic requirements**: The paper fits within the two-page limit, and is adequately anonymized.

**Suggested Changes:**

This paper lies far outside my area of expertise, and my assessment of its correctness is therefore an educated guess. I apologize to the author and area chairs for my inability to offer useful comments.

---

### Official Review · Reviewer_kCAC · 2023-03-30

**Confidence:** 3

**Summary Of Contributions:**

The responsibility problem describes an issue when training neural networks with unordered targets: the fixed permutation of output units requires that each assume a responsibility for some elements. Prior work has proved the existence of the issue by identifying a single discontinuity. In this paper, the authors show that discontinuities under such models are uncountably infinite, motivating further research into neural networks for unordered data.

**Rating:**

Clear, Correct, and Reproducible (CCR): a submission which meets the reviewing criteria

**Strengths And Weaknesses:**

The contribution of the paper is clearly presented, but needs some fixes regarding English and grammatical aspects. In the Appendix are all the necessary demonstrations relative to the theorems and prepositions stated. The article, therefore, turns out to be correct and reproducible from a theoretical point of view.

**Suggested Changes:**

It would have been nice to see one experimental result. Finally, the paper could provide information about the limitation of the proposed approach.

---

### Comment · Area_Chair_gVqe · 2023-06-06
**Final meta-review: Invite to archive**

This work meets the threshold for archival, contents the URM statement and is deanonymized

---

### Meta-Review · Area_Chair_gVqe · 2023-04-04

**Recommendation:** Invite to archive
**Confidence:** 2

**Metareview:**

Thank you for this piece of work proving that maps which assign set elements to a fixed ordering must contain infinite discontinuities. Previous work showed the existence of the so-called _responsibility problem_ can be proven by identifying a discontinuity in the mapping, here the authors extend this proving that such discontinuities must in fact be uncountably infinite.

Pros:
* Clear presentation.
* Contains necessary proofs.

Cons:
* This paper is not for a general audience and makes only minimal attempts to explain the relevance of the result to non domain-experts. This is not a criticism so much as an observation. It does limit the potential impact.


Unfortunately, overall the reviewers and meta-reviewers have low confidence in this area of machine learning. However, from a theoretical point of view, to the best of their ability the reviewer kCAC judges this piece of work correct and reproducible thus I have decided to recommend it as "invite for archive", in line with their rating.

**Summary:**

This piece of work proves that maps which assign set elements to a fixed ordering must contain infinite discontinuities. Previous work showed the existence of the so-called _responsibility problem_ can be proven by identifying a discontinuity in the mapping, here the authors extend this proving that such discontinuities must in fact be uncountably infinite. Reviewers judge this work to be clearly presented, correct and reproducible.

**Reason For Not Giving A Higher Recommendation:**

The impact of the result for future theoretical or analytical work is not fully expanded on, limiting the potential impact. The authors say it has implications for any work related to "set prediction, including object detection, point cloud generation, molecular graph generation, speech separation, and more" ...but how so.

**Reason For Not Giving A Lower Recommendation:**

It is a very clear paper and the results seem to be justified given the content.

---

### Decision · Program_Chairs · 2023-04-07

Invite to archive

---

> ### Author Response · Authors · 2023-05-31
> **Acceptance of invitation to archive**
>
> We thank the reviewers for their comments and invitation to archive our TinyPaper. We would like to opt-in for archival, and have uploaded a de-anonymized manuscript.